# Genome-Wide Identification and Expression Analysis of *TPS* Gene Family in *Liriodendron chinense*

**DOI:** 10.3390/genes14030770

**Published:** 2023-03-22

**Authors:** Zijian Cao, Qianxi Ma, Yuhao Weng, Jisen Shi, Jinhui Chen, Zhaodong Hao

**Affiliations:** Key Laboratory of Forest Genetics & Biotechnology of Ministry of Education, Co-Innovation Center for Sustainable Forestry in Southern China, Nanjing Forestry University, Nanjing 210037, China

**Keywords:** *TPS* gene family, *Liriodendron chinense*, gene expression

## Abstract

Terpenoids play a key role in plant growth and development, supporting resistance regulation and terpene synthase (TPS), which is the last link in the synthesis process of terpenoids. *Liriodendron chinense,* commonly called the Chinese tulip tree, is a rare and endangered tree species of the family Magnoliaceae. However, the genome-wide identification of the *TPS* gene family and its transcriptional responses to development and abiotic stress are still unclear. In the present study, we identified a total of 58 *TPS* genes throughout the *L. chinense* genome. A phylogenetic tree analysis showed that they were clustered into five subfamilies and unevenly distributed across six chromosomes. A cis-acting element analysis indicated that *LcTPSs* were assumed to be highly responsive to stress hormones, such as methyl jasmonate (MeJA) and abscisic acid (ABA). Consistent with this, transcriptome data showed that most *LcTPS* genes responded to abiotic stress, such as cold, drought, and hot stress, at the transcriptional level. Further analysis showed that *LcTPS* genes were expressed in a tissue-dependent manner, especially in buds, leaves, and bark. Quantitative reverse transcription PCR (qRT-PCR) analysis confirmed that *LcTPS* expression was significantly higher in mature leaves compared to young leaves. These results provide a reference for understanding the function and role of the *TPS* family, laying a foundation for further study of the regulation of *TPS* in terpenoid biosynthesis in *L. chinense*.

## 1. Introduction

Terpenoids, also known as isoprenoids, are a class of natural products with extremely rich structures. Currently, more than 80,000 terpenoids and their derivatives have been found in bacteria, fungi, insects, and plants [1]. These terpenes are composed of isoprene (C5) as the basic structural unit, which is spliced by two or more 5-carbon isoprene units [2]. The conventional terpenes are the head and tail condensation of isoprene units, and the head and tail condensation can occur in two ways, involving trans-isoprene diphosphate and cis-isoprene diphosphate. The condensation products of two isoprene units are called monoterpenes (with 10 carbon skeletons), the condensation products of three isoprene units are called sesquiterpenes (C15), and the condensation products of four isoprene units are called diterpenes (C20) [3].

Terpenoids exist in all organisms, but compared with other organisms, there are hundreds of terpenoids in green plants, especially in flowering plants, which play an important role in both the primary and secondary metabolism of the plants [3]. A small number of terpenoids participate in primary metabolism [4], such as hormones [5], components of the electron transport systems, protein modifiers, determinants of membrane fluidity, and antioxidants [6]. The terpenoids of primary metabolism have been shown to play important roles in sustaining life activities [7]. For example, carotenoids participate in photosynthetic reactions as photosynthetic pigments [8,9]; gibberellins, abscisic acid, and cytokinins participate as regulators of plant growth and development [10]; phytosterols participate as influencing factors of membrane structure and function [11,12]; ubiquinone and plastoquinone side chains function as electron carriers; and polyphenols participate in protein glycosylation [13]. However, terpenoids in most plants exist in the form of secondary metabolites, which do not directly participate in plant growth and development, but have certain physiological functions, such as plant defense responses [14], pharmacological compounds [15], and fragrance volatilization [4,16,17]. Terpenoids of secondary metabolism act as signal transduction in the interactions between plants and the environment, and between plants and animals, as well as between plants [18]. Many terpenoids show strong toxicity and are the core components to resist herbivores, pests, and microbial pathogens. Volatile terpenes, as floral components, play a role in pollinating plants [19]. Volatile terpenes can also serve as interspecific or interplant “alarm” signals to trigger the defense responses of neighboring plants [20]. Terpenes, such as artemisinin, are used to treat malaria [21].

Terpene synthase (TPS) is the key enzyme for sesquiterpene synthesis. In higher plants, many *TPS* genes exist in families. With the development of modern sequencing technologies, the *TPS* gene has attracted increasing attention. In Dendrobium officinale, a total of 34 *TPS* genes were identified using genomic and transcriptome data, and these genes were divided into four subfamilies, which were mainly expressed in the flowers, followed by the roots and stems [22]. In the genome of *Camellia*, a total of 80 TPS genes were identified, and most *CsTPS* genes were transcriptionally enhanced by methyl jasmonate [23]. Research exploring the response to abiotic stress has found that 26 *TPS* genes were identified in the aloes genome, they possessed many cis-acting elements related to stress, and plant hormones were discovered in their promoter regions [24]. Meanwhile, many of the *TPS* genes in roses were up-regulated in response to osmotic stress and heat stress [25]. In addition, many *TPS* genes have been reported in other higher plants or bacteria [26,27].

*Liriodendron*, an ancient relict genus, is comprised of two woody plants, i.e., *Liriodendron tulipifera* and *L. chinense* [28]. Among them, *L. chinesne* is naturally distributed in eastern Asia, while its vicarious species, *L. tulipifera*, grows in eastern North America, comprising a well-known classical intercontinental disjunct distribution [29,30]. *L. chinesne* is a rare and endangered tree species, while *L. tulipifera* is a dominant tree species in local forests. In addition, this pair of vicarious species has diverged over 10–16 million years [31], but they are still similar in general morphological and phenological characteristics, and interspecific hybridization is compatible, showing obvious heterosis. The completion of the *L. chinense* genome sequencing laid the foundation for studying the special role of *L. chinense* in angiosperm phylogeny [32], exploring the gene expression patterns of *L. chinense*, and utilizing the potential gene resources of *L. chinense*. The in-depth analysis of the *TPS* gene family in *L. chinense* by bioinformatics can provide a theoretical basis for the functional identification of *TPS* genes in *L. chinense*, providing a reference for exploring the stress resistance of *L. chinense*.

## 2. Materials and Methods

### 2.1. Identification of TPS Family Members in L. chinense

Genome and proteome data of *L. chinense* were downloaded from the *L. chinense* protein database (https://hardwoodgenomics.org/Genome-assembly/2630420 (accessed on 12 March 2022)) for subsequent identification and analysis. There are two specific conserved domains located at the C and N-terminal of TPS, respectively. The hidden Markov model (HMM) files of PF03936 (C-terminal) and PF01397 (N-terminal) were obtained from the Pfam database [33], which were used to retrieve candidate genes in HMMER (V3.3). The local BLAST search was performed based on the reference sequence. The Pfam results were combined with the BLAST search results; NCBI-BLAST and NCBI-CD-Search were used for confirmation, and manual correction was performed. The retrieved TPS proteins were validated using BLASTp to remove sequences that did not contain the C-terminal and N-terminal domains of TPS. The online software ExPASy ProtParam tool was used to analyze the physical and chemical properties of the LcTPS proteins, Compute pI was used to analyze the isoelectric points of the LcTPS proteins, and ProtParam was used to analyze the amino acid number, hydrophilicity, aliphatic index, molecular weight, and other information regarding the LcTPS proteins.

### 2.2. Conserved Motifs, Gene Structure, and Phylogenetic Analysis

The default parameters of the online software MEME [34] were used to analyze the conservative domains of the TPS proteins, and 10 conserved domains and motifs of the LcTPS proteins were obtained. The TPS protein sequences of *Populus euphratica, Selaginella moellendorffii, Arabidopsis thaliana, Amborella trichopoda, Abies grandis,* and *O. sativa* were obtained from a supplementary document for *TPS* studies of *D. officinale* [22]. MAGA 11 [35] software was used for multi-sequence comparison, and trimAl [36] was used for sequence shearing. The phylogenetic tree was constructed by the Neighbor-Joining (NJ) method, and all the parameters were default values, except that the check parameters were repeated 1000 times. The phylogenetic tree was visualized using iTOL [37].

### 2.3. Chromosome Mapping of the TPS Genes in L. chinense

According to the genetic information provided by the *L. chinense* protein database (https://hardwoodgenomics.org/Genome-assembly/2630420 (accessed on 12 March 2022)), the starting and ending positions of TPS genes in each *L. chinense* distribution region on the chromosomes were obtained, and the chromosome distribution map of genes was drawn, using TBtools [38] software, in proportion to the length of the related genes and chromosomes.

### 2.4. Cis-Acting Element Analysis of the TPS Genes in L. chinense

A sequence of 2000 bp upstream of the *LcTPS* promoter was extracted, and the cis-acting element prediction was performed on the obtained sequence, using the online software PlantCARE [39], and collated and counted.

### 2.5. Synteny Analysis of the TPS Gene Family in L. chinense

The One Step MC ScanX [40] in TBtools was used to calculate the Synteny analysis of 6 species, including *L. chinense, Cycas revoluta, O. sativa, A. trichopoda, Populus trichocarpa,* and *A. thaliana*, after which the synteny analysis files were visualized by the Dual Synteny Plot for MC ScanX. The synteny analysis relationship between *L. chinense* and the other five species was obtained.

### 2.6. Gene Expression Analysis Based on Transcriptome Data

To understand the dynamic expression patterns of the *LcTPS* family genes, transcriptome data from 8 different tissue sites (xylem, phloem, stamen, stigmas, sepals, leaves, buds, and bark) and 3 abiotic stresses (low temperature, high temperature, and drought stress) were used to remove non-expression and low expression levels. The Heatmap [41] package was used to draw the heatmap of *LcTPS* gene expression and analyze its expression pattern. The transcriptome data regarding cold and heat stress (PRJNA679089) and drought stress (PRJNA679101) was downloaded from NCBI. We analyzed the transcriptome using methods described previously [42].

### 2.7. Plant Materials for RT-qPCR Analysis of the L. chinense TPS Gene

The plant materials used in this experiment came from *L. chinense* grown on the campus of Nanjing Forestry University. Four kinds of tissue materials, including young leaves, mature leaves, non-lignified stems, and lignified stems, were collected in May. After collection, they were immediately placed in liquid nitrogen and then stored in the refrigerator at −80 °C in the laboratory. RNA was extracted using an RNA-extraction kit (Promega, Shanghai, China). RNA is reversed into cDNA using the Hiscript III 1st Strand cDNA Synthesis Kit (Vazyme, Nanjing, China). qRT-PCR is performed using a LightCycler 480II (Roche, Basel, Switzerland) with the AceQ qPCR SYBR Green Master Mix (Vazyme, Nanjing, China). We used three biological replicates and three technical replicates to perform qRT-PCR. The relative expression levels of the genes were used for gene expression analysis. For qRT-PCR data, the experimental results were analyzed by the 2^−ΔΔCt^ method (Livak method) [43]. The expression level of *Lchi16533* gene in young leaves was set to 1. A qRT-PCR analysis was used to confirm the expression patterns of ten genes (*Lchi29758, Lchi25750, Lchi30655, Lchi25434, Lchi16533, Lchi17170, Lchi29760, Lchi32068, Lchi26814,* and *Lchi29762*) in different tissues (Appendix A).

## 3. Results

### 3.1. Genome-Wide Identification and Basic Information of the L. chinense TPS Proteins

TPSs constitute a moderate-sized gene family, and the number varies greatly from plant to plant. So far, the *TPS* family has been identified in many plants. Each TPS has two conserved domains, PF01397 (N-terminal domain of terpenoid synthase) and PF03936 (metal ion-binding domain) [44]. First, the PF03936 and PF01397 domain model data in the Pfam database were used as models, and HMM was used to retrieve the candidate genes. A total of 63 candidate TPS proteins were obtained from the PF01397 domain, and 80 candidate TPS proteins were obtained from the PF03936 domain. To verify the sequences, BLASTp searched the retrieved TPS proteins and removed the sequences that did not contain the C-terminal and N-terminal domains of TPS, resulting in 82 candidate TPS proteins. Subsequently, TPS protein sequences from 8 species were downloaded from Phytozome and were BLAST-searched and confirmed by NCBI-blast and NCBI -CD-Search, resulting in 68 candidate TPS proteins. Finally, a total of 58 LcTPS protein sequences were obtained by the intersection of Pfam results and BLAST search results.

Using the online tools provided by ExPASy, the basic information of each protein was counted. Information including isoelectric point, molecular weight, number of amino acids, fat coefficient, and mean hydrophilicity and hydrophobicity were included, and the subfamily of each gene was marked (Appendix A). The results showed that the number of amino acids in *L. chinense* ranged from 95 (*Lchi32142*) to 983 (*Lchi29766*). The molecular mass ranged from 10,946.67 Da (*Lchi32142*) to 113,479.69 Da (*Lchi297660*). The grand average of the hydropathicity (GRAVY) of LcTPS was negative, except for *Lchi25431*, indicating that most LcTPSs were hydrophilic proteins. The aliphatic index (AI) of the LcTPS proteins ranged from 76.74 (*Lchi29754*) to 104.66 (*Lchi25431*). Most isoelectric points (pI) were between 4.53 (*Lchi25431*) and 6.41 (*Lchi29754*), with the exception of *Lchi30652* (8.68) and *Lchi32142* (8.88), which had higher isoelectric points. It is hypothesized that most *L. chinense TPS* genes encode weakly acidic proteins and function in acidic subcellular environments.

### 3.2. Analysis of the Conserved Motif, Gene Structure, and Phylogeny

The selected 58 LcTPS proteins were compared with *TPS-a, TPS-b, TPS-c, TPS-d, TPS-g,* and *TPS-e/f* of *P. trichocarpa* (32 TPSs), *S. Moellendorffii* (14 TPSs), *A. thaliana* (32 TPSs), *A. trichopoda* (13 TPSs), *A. grandis* (11 TPSs), and *O. sativa* (33 TPSs). The conserved sequences of all members of the *TPS-e/f* and *TPS-h* subfamilies were used to construct the NJ phylogenetic tree (Figure 1). The plant *TPS* family can be divided into seven subfamilies. The *TPS-h* subfamily is unique to *Selaginella tamariscina*, and *TPS-d* is unique to gymnosperms. According to the cluster analysis, *LcTPSs* were distributed in the other five subfamilies, including 23, 17, 2, 6, and 10 LcTPS proteins in the *TPS-a, TPS-b, TPS-c, TPS-g,* and *TPS-e/f* subfamilies, respectively. The *TPS-a* subfamily has the largest number of LcTPS proteins, which is in line with the previous analysis that the *TPS-a* subfamily is the largest branch of *TPSs*, indicating that the *L. chinense TPS* genes mainly synthesize sesquiterpenes and diterpenes. *TPS-c* and *TPS-e/f,* which are less abundant in *L. chinense,* are subfamilies shared by angiosperms and gymnosperms,.

To further elucidate the structural and functional characteristics of *L. chinense* TPS proteins, the conserved motifs of the LcTPS proteins were identified using MEME software (Figure 2 and Appendix A). We found that motifs 2 and 10 had the highest and lowest frequency, respectively. When the LcTPS proteins were sorted according to subfamily classification, it was found that the TPS-a subfamily had the most proteins with all the conserved motifs (30.43%), while proteins in the three subfamilies of *TPS-c, TPS-e/f,* and *TPS-g* had lost at least one of these conserved motifs, indicating that the proteins of these three subfamilies may have undergone extensive functional differentiation.

### 3.3. Chromosomal Distribution and Collinearity Analysis of the TPS Gene Family in L. chinense

The chromosomal mapping showed that 58 *LcTPS* gene family members were mainly located on six chromosomes of *L. chinense* (Figure 3). The analysis found that the number of *TPS* gene sequences located on chromosome 1 was the largest (20), followed by chromosome 19 (14). Four were distributed on chromosome 3, two were distributed on chromosome 12, and two genes were evenly distributed on chromosomes 4 and 6. Chromosomal mapping also showed that the 58 *L. chinense* TPS genes were mostly distributed at both ends of the chromosome. Considering that the genes on the chromosome generally show a cluster distribution, we suspect that many *TPSs* originated from tandem or segmental duplications.

Fifty-eight TPS genes of *L. chinense* were used for collinearity analysis with *TPS* genes of the remaining species. There were no *TPS* genes in the collinear block of *L. chinense* and *C. revoluta*, but there was a collinear relationship between *L. chinense* and the *TPS* genes of *A. trichopoda* and *P. trichocarpa*, which are also trees. Since there are no chromosomal-level genomic data for *A. trichopoda*, it is possible that more collinearity is masked. In Appendix A, *L. chinense* has one pair of genes located in the collinearity region with *O. sativa*, three pairs with *A. thaliana*, four pairs with *A. trichopoda*, and seven pairs with *P. trichocarpa*. Notably, the collinear regions were concentrated on chromosomes 1 and 19 of *L. chinense*. Combined with the chromosomal mapping results, more *TPS* genes were located on chromosome 1 and chromosome 19. The only gene with a collinear relationship between O. sativa and *L. chinense* was *Lchi21269* on chromosome 19. The collinear relationship between *A. thaliana* and *L. chinense* was found to be based on two genes, *Lchi16512* and *Lchi12906*, on chromosome 1, and *Lchi25434* on chromosome 19. The genes with a collinear relationship between *A. trichopoda* and *L. chinense* were *Lchi16512* on chromosome 1, *Lchi12048* on chromosome 6, and two genes, *Lchi25434* and *Lchi21269,* on chromosome 19. The genes with a collinear relationship between *P. trichocarpa* and *L. chinense* are as follows: three genes, *Lchi16511, Lchi16521,* and *Lchi1653,* on chromosome 1; *Lchi12048* on chromosome 6; and three genes, *Lchi25434, Lchi32068,* and *Lchi23984*, on chromosome 19.

### 3.4. Transcriptional Responses of L. chinense TPS Genes to Abiotic Stress

To determine the potential biological role of the *LcTPS* genes in *L. chinense*, the 2000 bp region upstream of its initiation coding (ATG) was identified using the PlantCARE tool. In addition to a large number of core promoters (TATA-box) and enhancer elements (CAAT-box), we identified 19 cis-acting elements (Appendix A). MeJA responsiveness (21%) and abscisic acid responsiveness (22%) were found to have the highest amounts of cis-acting components (Figure 4a). *Lchi23979, Lchi23984,* and *Lchi29635* promoters contain cis-acting elements that are strongly related to MeJA responsiveness and abscisic acid responsiveness. These data suggest that *LcTPSs* may be relevant to MeJA responsiveness and abscisic acid responsiveness to participate in abiotic plant stress.

In order to study the expression level of the *LcTPS* genes under low temperature stress, the transcript data of the *L. chinense TPS* genes under low temperature stress were analyzed to explore the expression level of the *LcTPS* genes at 0 h, 12 h, 24 h, and 48 h after cold stress (Figure 4b). Among these, 24 *LcTPS* genes were not expressed or had low expression levels at different time points of control or low temperature treatment. In addition, 7 *LcTPS* genes were expressed in the control group, but the expression began to be down-regulated after low temperature treatment. The expression of most *LcTPS* genes was high at 12 h or 24 h after the low temperature treatment, and then gradually decreased to a low level. The expression of the *Lchi32250* gene was noticeable, among the genes, as it was immediately down-regulated by the low temperature treatment. However, among the genes whose expression level increased at 12 h after low temperature treatment, *Lchi30655* had the highest absolute expression. It is worth noting that the expression level of *Lchi30655* decreased at 24 h, but increased at 48 h and was the highest at 48 h.

In order to research the expression level of the *LcTPS* genes under a high temperature stress, the transcript data of the *L. chinense TPS* genes under high temperature stress were analyzed to explore the expression levels of the *LcTPS* genes at 0 h, 1 h, 3 h, 6 h, 12 h, 24 h, and 72 h after high temperature stress (Figure 4c). Among them, 16 *LcTPS* genes were not expressed or had low expression levels at different time points after control or high temperature treatment. In addition, the expression levels of most of the *LcTPS* genes were the highest in the control group, but gradually decreased after the high temperature stress treatment. Two of these genes (*Lchi32250* and *Lchi14126*) showed a trend of first decreasing, then increasing, and then decreasing after high temperature stress treatment. In addition, some *LcTPS* genes were specifically highly expressed at 1 h, 6 h, 12 h, and 72 h after high temperature treatment, and most of these genes showed very low levels in the control, indicating that these *LcTPS* genes may play a role in the response of *L. chinense* to high temperature stress at different time points after high temperature treatment.

In order to study the expression level of the *LcTPS* genes under drought stress, the transcript data of the *L. chinense TPS* genes under drought stress were analyzed to explore the expression level of the *LcTPS* genes at 0 h, 1 h, 3 h, 6 h, 12 h, 24 h, and 72 h after drought stress (Figure 4d). Among them, 13 *LcTPS* genes were not expressed or had low expression levels at different time points after the control or drought treatment. In addition, the expression level of most of the *LcTPS* genes reached the highest level at a certain time point after drought treatment, and the expression level was low or not expressed in the control, indicating that these *LcTPS* genes may act on the drought stress response of *L. chinense* at different time points after drought treatment.

### 3.5. An Expression Analysis of L. chinense TPS Genes across Different Tissues

To investigate the role of *LcTPS* genes in *L. chinense* growth and development, we analyzed *L. chinense* tissue transcript data, including eight tissue sites (phloem, stigma, xylem, bud, stamen, leaf bark, and sepal). In addition to *Lchi21906, Lchi16521, Lchi32745,* and *Lchi29754*, four genes were not expressed at all in the eight tissue sites, and *Lchi24760* was expressed at a low level in each tissue site. The other genes were highly expressed in one or more tissues and organs to varying degrees (Figure 5a). *Lchi30652, Lchi30424, Lchi23979,* and *Lchi21274* genes were specifically expressed in xylem and phloem, especially in xylem. It is speculated that phloem and xylem could not be easily distinguished during sampling, resulting in the contamination of xylem tissue in phloem samples. Few *LcTPS* genes were specifically expressed in flower organs, four genes were specifically expressed in stigma and sepal, and only one gene was expressed relatively highly in stamens, but this gene was also expressed relatively highly in bark. In addition, a large number of *LcTPS* genes were specifically expressed in buds, leaves, and bark, and there was a partial overlap between buds and leaves, suggesting that these genes may play a role in the biosynthesis of similar terpenoids in buds and leaves of *L. chinense*. In general, *LcTPS* genes showed a certain specificity in different tissues, and the number of *LcTPS* genes specifically expressed in vegetative organs was higher than that in reproductive organs, suggesting that *LcTPS* genes may play a role in the diversity of terpenoids during the development of vegetative and reproductive organs of *L. chinense*. At the same time, considering that the reproductive organs of *L. chinense* only last for approximately one month in a year, it is possible that the more specific expressions of the *LcTPS* genes in vegetative organs may contribute to the synthesis of more kinds of terpenoids, thereby promoting the biological resistance of vegetative organs such as the bark, leaves, and buds of *L. chinense* during the growth phase.

According to transcriptome data, the expression of *LcTPSs* in different tissues was significantly different. The results of the expression analysis showed that the expression level of the *L. chinense TPS* genes was higher in leaves and branches. Therefore, a quantitative real-time PCR was used to determine *Lchi16533, Lchi17170*, *Lchi25434*, *Lchi25750*, *Lchi26814*, *Lchi29758*, *Lchi29760*, *Lchi29762*, *Lchi30655*, and *Lchi32068*, respectively. The expression levels of 10 genes in young leaves, mature leaves, and non-lignified and lignified stems is shown in Figure 5b. When the expression level of the *Lchi16533* gene in young leaves was set to 1, the analysis results showed that the expression level of *Lchi29762* in non-lignified stems and mature leaves was much higher than that of other genes, while the expression levels of *Lchi29758* and *Lchi30655* genes in leaves and stems were the lowest compared with other genes. The expression of *Lchi16533* was the highest in mature leaves and the lowest in lignified stems. However, the significance between non-lignified stems and lignified stems was not obvious. The expression of *Lchi17170* was the highest in the mature leaves, and the expression of *Lchi17170* was low in the other three measured parts, with no significant difference between the mature leaves and the other three parts. The expression level of *Lchi25434* was the highest in the non-lignified stem and low in the other three parts; however, between the young leaves and the non-lignified stem, there was no obvious difference. The expression level of *Lchi25750* was the highest in young leaves, followed by lignified stems, but there was no significant difference between the four parts; however, the difference between the two parts was significant. The expression level of *Lchi26814* was the highest in mature leaves and the lowest in lignified stems. There was no significant difference in *Lchi26814* expression between non-lignified stems and mature leaves. The expression level of *Lchi29758* was the highest in lignified stems and the lowest in non-lignified stems, and the significance between young leaves and mature leaves was not obvious. The expression level of *Lchi29760* was the highest in mature leaves and the lowest in lignified stems; however, between young leaves and non-lignified stems, there was no difference. The expression of *Lchi29762* was high in both non-lignified stems and mature leaves, but not between the non-lignified stems and the other three parts. The expression level of *Lchi30655* was high in young leaves, but absent in the other three parts. The expression level of *Lchi32068* was the highest in mature leaves, lower at similar levels in non-lignified stems and lignified stems, and lowest in young leaves; however, between young leaves and lignified stems there was no significant difference.

## 4. Discussion

Terpenoids are important compounds to improve plant resistance, and there are many terpenoids in *L. chinense*. Terpene synthases function in the structural diversity of natural terpenoids by converting prenyl diphosphates into volatile monoterpenes and sesquiterpenes or semi-volatile and non-volatile diterpenes. Therefore, it is very important to study the terpenoid synthase gene family in *L. chinense*.

### 4.1. Genome-Wide Identification of TPS Genes in the L. chinense Genome

There were 58 *LcTPS* genes identified in the *L. chinense* genome based on conserved domains. After phylogenetic tree construction, the 58 *LcTPS* genes were divided into five subfamilies: *TPS-a, TPS-b, TPS-c, TPS-e/f,* and *TPS-g*. Among the *TPS* genes of *L. chinense*, there were no members of the *TPS-d* and *TPS-h* subfamilies. Some studies have found that *TPS-d* is an endemic subfamily of gymnosperms, and *TPS-h* is an endemic subfamily of Selaginella [45]. Our results are consistent with these findings. Our phylogenetic analysis of the *LcTPS* gene family revealed that *TPS-a* was the largest subfamily, with 23 genes, and *TPS-b* was the second largest subfamily, with 18 genes. This conclusion is consistent with findings in other plants. For example, in Solanum lycopersicum, the largest subfamily of *TPS* genes is *TPS-a* [46], and in *A. thaliana*, 22 out of 32 *TPS* gene families are *TPS-a* [47].

Terrestrial plants generally have a medium-sized *TPS* gene family formed by gene duplication [45]. We carried out chromosomal localization of *LcTPS* genes and found that some genes would form homologous clusters, which may be caused by gene replication events. Interestingly, we found 34 *LcTPS* genes distributed on chromosomes 1 and 19. Many plant *TPS* genes have highly conserved gene structures [47,48,49,50]; however, for *LcTPS* genes, this is not the case (Appendix A). The intron size of the *LcTPS* gene family is inconsistent. Some genes have only one intron, such as *Lchi30652*, and some genes have 12 introns, such as *Lchi29766*. The structure of *LcTPS* genes also varies greatly in intron length. For example, *Lchi27241* has a very long intron, while *Lchi12048* has a very short intron. ABA and MeJA are the main signal molecules affecting plant growth and development and stress response [51]. Through cis-acting element analysis of *LcTPS*, we found that *LcTPSs* were basically related to ABA and MeJA responses, so we speculated that *LcTPS* gene expression might be related to plant resistance (Figure 4a and Appendix A). In our analysis of the TPS gene family promoter cis-acting element, we found that genetic responses from *Lchi21274*, *Lchi29635,* and *Lchi29758* to ABA and MeJA are frequent. Under conditions of low temperature stress, we found that *Lchi212174* gene expression decreased with increasing exposure time, *Lchi29635* gene expression increased with increasing exposure time, and *Lchi29758* gene expression initially decreased and then increased with increasing exposure time. Under high temperature stress conditions, we found that the gene expression level of *Lchi29635* changed to 0 after one hour at a high temperature, while the gene expression level of *Lchi29758* decreased with a longer exposure period. Under drought stress, we found that the expression level of *Lchi21274* was basically unchanged, while the expression level of the *Lchi29635* gene was down-regulated with the increase in exposure time. Therefore, we suggest that *Lchi21274, Lchi29635,* and *Lchi29758* may respond to various stresses by regulating ABA and MeJA.

### 4.2. Transcriptional Responses of LcTPS Genes to Abiotic Stress and Tissue Development

By analyzing the expression of *LcTPS* genes under tissue-specific and abiotic stress, it was found that there was no uniform expression of *LcTPSs* under low temperatures, high temperatures, or drought stress, and that the expression level of some genes gradually decreased after stress (Figure 2 and Figure 4a–d). At the same time, the expression of many *LcTPS* genes was induced at different time points after stress treatment, but the expressed *LcTPS* genes were not the same in different treatments and at different time points after treatment, indicating that *L. chinense* may adopt different terpenoids in response to various abiotic stresses. At the same time, different terpenoids were used to respond to stress at different time points. Based on the tissue-specific expression results, four tissues and organs of *L. chinense* (young leaves, mature leaves, non-lignified stems, and lignified stems) were further selected for RT-qPCR analysis. The results showed that the expression levels of *Lchi29762* in non-lignified stems and mature leaves were much higher than those of other genes, while the expression levels of *Lchi29758* and *Lchi30655* genes in leaves and stems were lower than those of other genes (Figure 2). In summary, the expression of each gene in mature leaves and non-lignified stems is higher than that in the other parts. It can be speculated that *LcTPS* genes are mainly expressed in non-lignified stems and mature leaves, the functions of synthesized terpenoids are mainly reflected in leaves and stems, and volatile terpenoids are released on leaves to resist the effects of insects. The results of the present study lay a foundation for further verification of the ecological function of other *LcTPS* genes.

## 5. Conclusions

This paper reports the identification of 58 *LcTPS* genes in *L. chinense*, including 23 genes in the *TPS-a* subfamily, 17 genes in the *TPS-b* subfamily, 2 genes in the *TPS-c* subfamily, 6 genes in the *TPS-g* subfamily, and 20 genes in the *TPS-e/f* subfamily. Phylogenetic analysis, conserved motif analysis, gene structure analysis, and chromosomal localization were performed. The RNA-seq and RT-qPCR data analyses revealed that *LcTPSs* show specific expression patterns in different plant parts and under different stresses. In summary, this study provides a basis for further investigating the genetic and functional characteristics of the *LcTPS* gene family.

## Figures and Tables

**Figure 1 genes-14-00770-f001:**
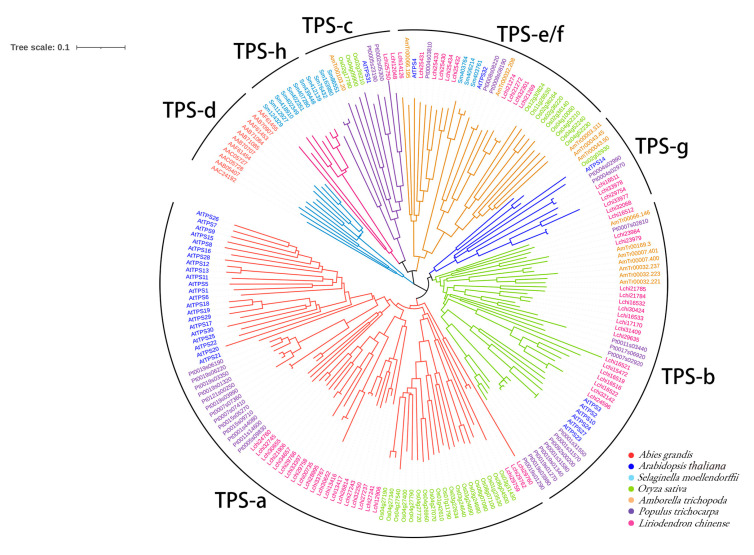
Phylogenetic analysis of Terpen synthase in *P. trichocarpa*, *S. Moellendorffii*, *A. thaliana*, *A. trichopoda*, *A. grandi*, *O. sativa*, and *L. chinense*.

**Figure 2 genes-14-00770-f002:**
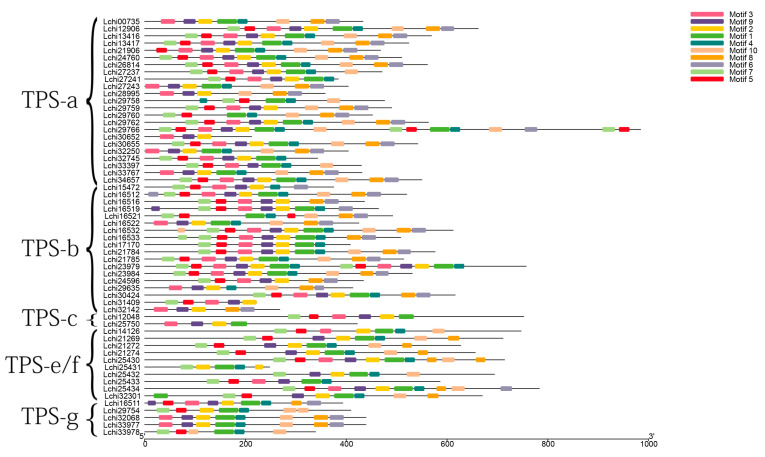
Conserved protein motifs from *L. chinense*. The numbers 1–10 of motif composition are displayed in different colored boxes in the LcTPS proteins.

**Figure 3 genes-14-00770-f003:**
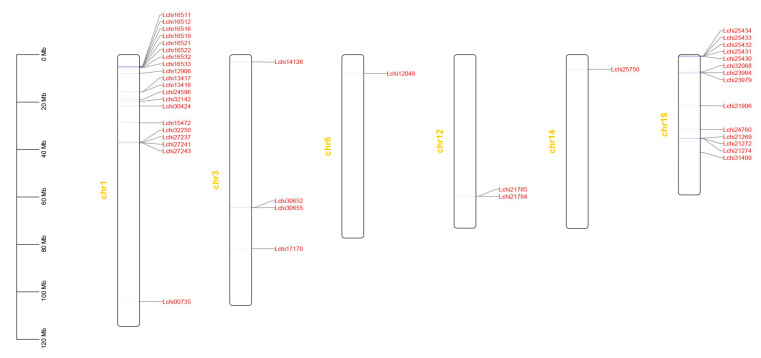
Distribution of the Terpen synthase genes on *L. chinense* chromosomes. The scale indicates a megabase (Mb).

**Figure 4 genes-14-00770-f004:**
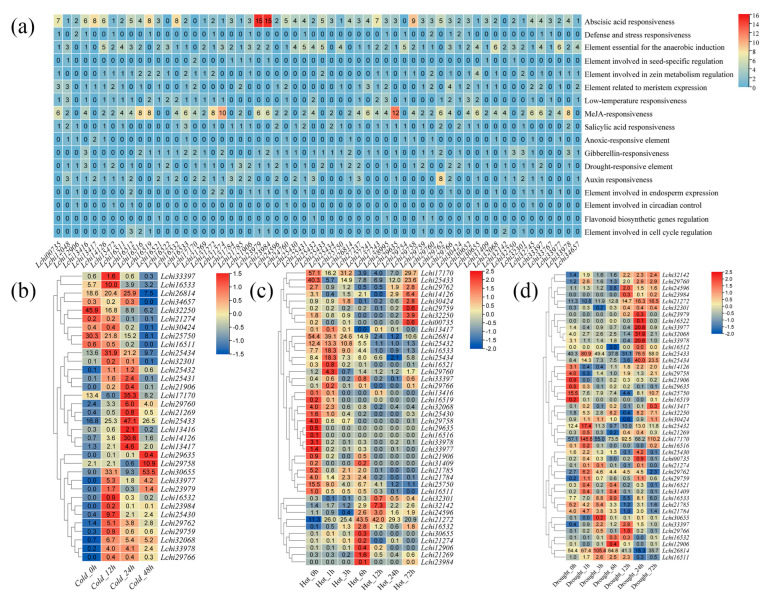
A cis-acting element analysis and the transcriptional responses of *TPS* genes in *L. chinense*. (**a**) The cis-acting elements were identified in the promoter regions of the *LcTPS* genes. The gradient color from blue to red, and the numbers in the grid, indicate the number of different cis-acting elements in *LcTPSs*. The transcription levels of *LcTPS* genes under low temperature (**b**), high temperature (**c**), and drought stress (**d**). The expression levels of the genes were represented by transcripts per million (TPM). The raw data of relative expression values and standard errors are provided in Appendix A.

**Figure 5 genes-14-00770-f005:**
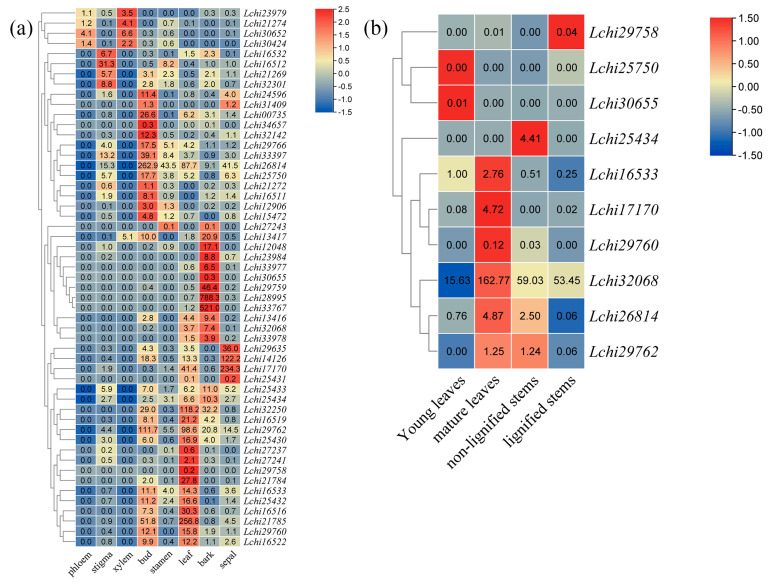
The expression dynamics of *LcTPS* genes across different tissues. (**a**) The expression levels of *LcTPS* genes were extracted from transcriptome data and quantified using transcripts per million (TPM). (**b**) Expression levels of selected *LcTPS* genes in young leaves, mature leaves, non−lignified stems, and lignified stems by qRT−PCR. The mean expression value was calculated from three independent biological replicates in different samples. The raw data of relative expression values and standard errors are provided in Appendix A.

## Data Availability

All analyzed transcriptome data were published on the NCBI website; cold and heat stress accession number is PRJNA679089, and drought stress accession number is PRJNA679101.

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
