# Peer review of "Genome-Wide Identification and Expression Analysis of *TPS* Gene Family in *Liriodendron chinense"

_genes, 2023, doi:10.3390/genes14030770_

Round 1

Reviewer 1 Report

The manuscript ‘Genome-wide Identification and Expression Analysis of TPS Gene Family in Liriodendron chinense’ deals with the identification of 58 TPS gene family members from this tree species. There is in depth and sound bioinformatics including homology based TPS gene identification, conserved motif analysis, gene structure and genomic location. The study is foundational for a better understanding of terpenoid biosynthesis governed by the TPS gene family in this species .

The experimental part needs clarification and elaboration in terms of the extent of data generated and analyzed in house as against the analysis of available genomic/ transcriptomic resources. The manuscript needs to be thoroughly checked for English grammar.

The specific comments are included below.

1.     It is not clear if the authors have done RNA sequencing (for low/high temperature, drought and tissue types) or used the transcriptome data that was available previously. There are no methods included for RNA seq analysis. If the already existing data was analyzed, the reference or source need to be included to give credit.

2.     In the materials section section 2.7, there is no mention of the RNA isolation procedure, that need to be included with information. Also include more details on qRT-PCR methods, biological and technical replicates and analysis.

3.     In the abstract line 15 ‘has high application value in regarding to forest chemicals’ need correction.

4.     The sentence in lines 350-352 ‘ The synthesis of terpenoids is derived from the conversion of monoterpenoids by terpenoid synthases, and the terpenoid synthase gene family controls the synthesis of these terpenoid synthases’ is ambiguous and need to be corrected.  

There are several such instances that can be improved for clarity.

Reviewer 2 Report

1- Add new information about the L. chinense plant in the introduction.

2- Use the full name of L. chinense in line 73.
3- use high-quality figures and pictures

4- Which method used for gene expression analysis?

5- arrange the references based on the journal format.

Round 2

Reviewer 1 Report

The authors have made all the recommended edits. However in the section 2.7, they can reduce the procedure details to include only the steps different from the manufacturer's instructions. 

Reviewer 2 Report

Accepted.